# Recent Advances in Extracellular Vesicles in Amyotrophic Lateral Sclerosis and Emergent Perspectives

**DOI:** 10.3390/cells12131763

**Published:** 2023-07-01

**Authors:** Gonçalo J. M. Afonso, Carla Cavaleiro, Jorge Valero, Sandra I. Mota, Elisabete Ferreiro

**Affiliations:** 1CNC-Center for Neuroscience and Cell Biology, University of Coimbra, 3004-504 Coimbra, Portugal; gjmafonso@cnc.uc.pt (G.J.M.A.); carla.cavaleiro@cnc.uc.pt (C.C.); 2Center for Innovative Biomedicine and Biotechnology, University of Coimbra, 3004-504 Coimbra, Portugal; 3III-Institute of Interdisciplinary Research, University of Coimbra, 3030-789 Coimbra, Portugal; 4Instituto de Neurociencias de Castilla y León, University of Salamanca, 37007 Salamanca, Spain; jorgevalero@usal.es; 5Institute of Biomedical Research of Salamanca (IBSAL), 37007 Salamanca, Spain; 6Department of Cell Biology and Pathology, University of Salamanca, 37007 Salamanca, Spain

**Keywords:** amyotrophic lateral sclerosis, neurodegenerative diseases, extracellular vesicles, exosomes, miRNA, biomarkers

## Abstract

Amyotrophic lateral sclerosis (ALS) is a severe and incurable neurodegenerative disease characterized by the progressive death of motor neurons, leading to paralysis and death. It is a rare disease characterized by high patient-to-patient heterogeneity, which makes its study arduous and complex. Extracellular vesicles (EVs) have emerged as important players in the development of ALS. Thus, ALS phenotype-expressing cells can spread their abnormal bioactive cargo through the secretion of EVs, even in distant tissues. Importantly, owing to their nature and composition, EVs’ formation and cargo can be exploited for better comprehension of this elusive disease and identification of novel biomarkers, as well as for potential therapeutic applications, such as those based on stem cell-derived exosomes. This review highlights recent advances in the identification of the role of EVs in ALS etiopathology and how EVs can be promising new therapeutic strategies.

## 1. The Current State-of-the-Art Research of ALS

Amyotrophic lateral sclerosis (ALS) is a fatal neurodegenerative disease. Described for the first time in the 19th century by Charcot, ALS is characterized by the degeneration of lower (spinal and bulbar) and upper (corticospinal) motor neurons [1]. The selective loss of motor neurons (MNs) in the primary motor cortex, brainstem, and spinal cord progressively leads to severe effects, such as loss of motor control, paralysis, and death. Death usually occurs due to respiratory failure. Approximately half of the patients with ALS show impairments in cognitive function and behavior, with 5–25% of patients developing frontotemporal dementia (FTD) [2,3,4], which is an uncommon type of dementia characterized by changes in the frontal and temporal lobes.

ALS is considered a rare disease, with an incidence that ranges between 0.6 and 3.8 persons out of 100,000 and a prevalence between 4.1 and 8.4 per 100,000 individuals, with an average age of onset between 51 and 66 years [5]. The life expectancy of patients with ALS is short, ranging between 24 and 50 months. However, approximately 10% of patients manage to live for more than 10 years [5], a fact that reflects the high patient-to-patient phenotypic variability that characterizes ALS. The triggering elements of the disease remain unknown, although genetic causes can be attributed to several individuals. Some studies point to the possibility of an oligogenic or polygenic nature, as mutations in two or more genes may be required for the disease [6,7]. People with a history of ALS in their family and those carrying ALS-related genes are more likely to develop the disease (familial, fALS), representing 5–10% of all cases. For the remaining 90% to 95%, the illness can occur spontaneously, without a family history (sporadic ALS) [6], and still be linked to ALS-related genes. Currently, ALS is difficult to diagnose due to the absence of a test that can solely lead to its identification unless it is a familial form. In the absence of a family history, a battery of examinations is often performed to exclude other possible pathologies. Currently, ALS remains cureless, and the available treatments are sparse and mostly palliative. Two approved medications are currently prescribed to patients, Riluzole and Edaravone, with the latter only being approved in some countries. However, they only present small benefits in delaying ALS progression, usually only by a few months [8]. Therefore, the discovery of new and effective drugs is of utmost importance.

### 1.1. Risk Factors for ALS Onset and Progression

The likelihood of developing ALS and its progression are influenced by numerous factors, including genetic and non-genetic origins. One important non-genetic factor is age, as individuals who develop ALS in early adulthood tend to experience slower disease progression rates [9,10]. Another factor is gender, with men being about 1.3 times more likely to develop ALS than women and earlier in life [11]. Gender also plays a role in ALS onset type, with spinal onset more common in men, while women are more likely to present bulbar onset [10]. In addition to genetic factors, exposure to certain modifiers throughout an individual’s life may also contribute to the risk of developing ALS [12]. Several environmental and lifestyle factors have been identified as potential risk factors for ALS onset, including hazardous smoking habits [13], higher lipid levels [14], prolonged exposure to pollutants [15], heavy metals [16], chemicals [17], electromagnetic fields [17], a history of electric shock [18], and head trauma [19]. Other factors linked to an increased risk of ALS include military service [20], participation in professional sports [21,22], and occupations that involve repetitive physical work [12,23,24]. However, some of these factors have been contested, owing to studies with inconclusive results [25]. These factors can eventually lead to epigenetic and genomic changes that may contribute to ALS onset, such as the occurrence of *C9ORF72* (chromosome 9 open reading frame 72) somatic mutations [26]. Scientific reports have consistently indicated an interaction between the genetic and environmental risk factors. Epigenetic alterations, mostly comprising DNA methylation, were identified by screening the biofluids and postmortem brain and spinal cord tissues. In this regard, Morahan et al. (2009) reported gene and CpG island methylation in 38 differentially methylated sites in brain samples from 10 sALS patients [27]. Similarly, Figueroa-Romero et al. (2012) identified 3574 methylated genes in postmortem sALS patients’ spinal cords [28]. Cai et al. (2022) recently proposed a role for DNA methylation in ALS pathogenesis. Their study analyzed and compared the blood of 32 healthy controls with 32 sALS patients, leading to the identification of 12 differentially methylated regions (DMRs) in 12 genes and 34 differentially methylated positions (DMPs) in 13 genes. Abnormal methylation patterns were primarily associated with genes involved in the regulation of crucial cellular functions that have previously been linked to ALS, including microtubule-based movement, ATP-nucleotide binding, and neuronal apoptosis [29]. Despite research efforts to elucidate the impact of environmental and lifestyle factors on different cellular and molecular processes involved in ALS onset and progression, the exact mechanisms underlying motor neuron degeneration are still not sufficiently understood [1].

### 1.2. ALS Genetics and Associated Mechanisms

ALS is a highly heterogeneous disease caused by a wide array of different genes with hundreds of possible mutations [30]. Consequently, distinct fundamental cellular processes have been reported to be dysfunctional in different stages of the disease, including DNA repair mechanisms, RNA metabolism, mRNA axonal transport, protein homeostasis, protein trafficking, protein misfolding and aggregation, calcium regulation, mitochondrial function [31], redox signaling, lipid metabolism, glutamate signaling, and autophagy [32]. ALS-related gene mutations may also affect intercellular communication and function, such as neurovascular function [33,34,35], glial-related neuroinflammation [10,36,37,38], and neuron–glia interaction [39,40]. Among the several genes identified as ALS-related, some are involved in both fALS and sALS, such as *TDP-43* (TAR DNA-binding protein 43), also known as *TARDBP* (transactive response DNA-binding protein), *SOD1* (copper zinc superoxide dismutase 1), *C9ORF72*, and *FUS* (fused in sarcoma), among others [41]. Nevertheless, for 32% and 89% of patients with fALS and sALS, respectively, the mutations involved are unknown [42,43].

One of the most studied ALS-related genes is SOD1, which encodes for an important antioxidant protein, superoxide dismutase [44], responsible for converting superoxide radicals in hydrogen peroxide and oxygen [45]. Mutant SOD1 (mSOD1) alters different metabolic pathways and results in the formation of misfolded SOD1 protein aggregates and neurodegeneration [46,47]. Accordingly, mSOD1 aggregate accumulation impairs axonal transport and is neurotoxic to spinal cord MNs from the pre-symptomatic phase onwards in the ALS mice SOD1-G93A model [48]. mSOD 1 is also responsible for the alteration of the dynamic interaction between MNs and their surrounding glial cells, evoking a non-cell autonomous toxicity mechanism driven either by the promotion of the secretion of neurotoxic cytokines, through the loss of glial cells supporting properties, or both, leading to the death of MNs [49,50]. In one proposed mechanism, extracellular mSOD1 is endocytosed by microglia and activates caspase-1, leading to the upregulation of IL-1β [51,52]. IL-1β is a proinflammatory cytokine that is potentially involved in ALS neuroinflammation-related processes [53], like microgliosis and astrogliosis. In postmortem tissue samples from ALS patients, microglia are in a proinflammatory state [54] and release several cytokines, such as IL-1α and TNF-α, which induce astrocyte neurotoxicity [54]. Such evidence points to deleterious crosstalk between microglia and astrocytes, thus tracing an increased proinflammatory and neurotoxic microenvironment. Therefore, the progressive degeneration of corticospinal and spinal motor neurons may depend on their vulnerability to both mSOD1 aggregate accumulation and the effects of the surrounding glial cell dysregulation, which emphasizes the simultaneous occurrence of lower and upper MN degeneration [55].

The most commonly mutated gene in both patients with fALS and sALS is *C9ORF72*. The *C9ORF72* gene contains 11 exons, and (GGGGCC)n is located between exons 1a and 1b. (GGGGCC)n is located in the first intron of V1 and V3 and in the promoter region of variant 2. This gene codes for a protein with the same name whose function is not fully understood but is thought to be involved in different cellular activities, such as protein transport, vesicle formation, autophagy, RNA processing, and cell signaling [56,57]. It has been suggested that C9orf72 may play a role in the autosomal and lysosomal function of macrophages and microglia through the regulation of inflammatory responses, possibly related to MN survival, relevant in ALS [58,59]. Wild-type C9orf72 forms a complex with SMCR8 (Smith–Magenis syndrome chromosomal region candidate gene 8) and WDR41 (WD40 repeat-containing protein 41) to perform these functions, including their effect on macrophages and microglia [57,60]. Because of the nature of this procedure, this function has been proposed to be affected in ALS in the presence of mutations; however, further studies are needed [57,59]. *C9ORF72* mutation is not only the most common mutation in ALS but is also responsible for FTD. This mutation is reflected as an increase in the number of hexanucleotide (G4C2)n repeat expansions (HRE) in the noncoding region of *C9ORF72*, which results in both loss of function linked to *C9ORF72* haploinsufficiency and a gain of function, resulting in the expression of abnormal bidirectionally transcribed RNAs carrying the repeat [61]. This repeat expansion leads to abnormal RNA molecule biosynthesis, which is then translated into dipeptide repeat proteins (DPRs) containing multiple copies of the specific amino acid sequence GGGGCC. DPRs, such as poly-proline-arginine (poly-PR), poly-glycine-arginine (poly-GR), and poly-glycine-alanine (poly-GA), are cytotoxic [62], accumulate in neurons [63], and may then spread to glial cells via intercellular communication [64], thus impairing protein folding and transport, inducing oxidative stress, and disrupting mitochondrial function [65]. An important player in the pathophysiology of ALS patients carrying *C9ORF72* expansion is poly GA, which induces the intracellular aggregation of phosphorylated TDP-43 proteins through the impairment of TDP43 nuclear translocation and cytoplasmic mislocation [66,67]. A pathological hallmark of these patients is the presence of TDP-43 inclusions in neurons and oligodendroglial cells. The *C9ORF72* gene has also recently been associated with nucleolar dysfunction [68] and DNA repair inhibition [69]. Other important cellular processes that are affected by *C9ORF72* gene mutation are vesicular and protein trafficking [70]. *C9ORF72* HRE was found to reduce the interaction between C9orf72 and the Rab GTPase key regulator Rab7L1, resulting in decreased extracellular vesicle (EV) release [70]. The role of C9orf72 in protein trafficking was further demonstrated in the human spinal cord of an ALS patient (with a *C9ORF72*-intronic repeat expansion mutation), where an increased proportion of motor neurons showed the colocalization of C9orf72 with Rab 5, Rab 7, and Rab 11 (when compared to healthy individuals), possibly resulting in the dysregulation of endosomal trafficking [70]. Interestingly, these proteins were recognized to be associated with vesicle trafficking regulation from the multivesicular bodies (MVB) to the plasma membrane, being involved, among other instances, in autophagy [70].

*TARDBP*, which codes for TDP-43, is another commonly mutated gene in ALS. Under normal physiological conditions, TDP-43 is primarily found in the nucleus, where it participates in the regulation of gene expression [71]. However, mutations in this gene in ALS or FTD patients lead to mislocalization of the corresponding protein, accumulating in the cytoplasm in the form of abnormal TDP-43 aggregates and generating anomalous ubiquitin-positive inclusions in the nucleus and cytoplasm [72]. These inclusions can affect the physiological functions of p62 (also known as SQSTM), which is involved in autophagy and proteasome regulation. The sequestration of p62 within TDP-43 aggregates leads to the impairment of autophagy and proteasome functions, driving the further accumulation of misfolded proteins within cells [73,74]. Indeed, aggregates colocalizing TDP-43 with p62 and SOD1 were found in postmortem ventral spinal cord tissues of patients with fALS and sALS, despite the existence of different aggregation profiles [74]. This can occur even in the absence of mutations in the respective genes, which may be attributed to incorrect protein folding, namely SOD1 [74,75].

Another commonly mutated ALS-linked gene is *FUS*, which encodes the RNA-binding protein FUS. In healthy individuals, FUS is found in the nucleus and is related to gene expression regulation, DNA repair, and RNA processing [76]. However, in patients with ALS and FTD, FUS translocates into the cytoplasm, creating FUS inclusions that can boost further nefarious effects, such as RNA mislocation associated with sequestering of the motor protein kinesin-1 [77] and axonal transport impairments [78]. FUS mutations in ALS may also impair mitochondrial functions through the sequestration of respiratory chain complex mRNAs in the cytoplasm [79]. Moreover, FUS loss of function can lead to neuronal dysfunction and death [80]. It is possible that FUS mislocation into the cytoplasm may contribute to their incorporation into EVs, and then, by dissemination to other cells via intercellular transfer, the phenotype is spread into the circulation [81,82].

## 2. Extracellular Vesicles and Their Role in ALS Onset and Development

### 2.1. EVs Overview

Extracellular vesicles (EVs) are endogenous bilipid layers, plasma membranes, or endosome-derived nanoparticles released by most eukaryotic cells into the extracellular space [83]. They were first described by [84] and were initially thought to be cellular waste products. Most studies have reported that cells can synthesize and secrete three main types of EVs: exosomes (exosome-like vesicles), microvesicles or ectosomes, and apoptotic bodies [85,86]. However, more recently, other types of EVs, such as retrovirus-like vesicles and vesicles, have been reported. The former are 90–100 nm particles that possess a subset of retroviral proteins and carry endogenous retroviral sequences but not for cellular entry or retroviral propagation [87]. Mitovesicles are of mitochondrial origin, possessing components of this organelle such as mitochondrial proteins, lipids, and mitochondrial DNA (mtDNA) [88]. Mitovesicles are distinguishable from exosomes and microvesicles based on their morphology, size, and content [88]. EV classification relies on several parameters, such as size, content, function, biogenesis, and release pathways [85]. The biological functions of EVs depend on their type and highly specific bioactive cargo, which represent the progenitor cell state [89,90]. There are different ways to identify EVs, such as physical characterization through microscopy, proteomic analysis, RNA sequencing, functional characterization, and biochemical analysis of their composition [91]. An important way to identify EVs is through the presence of specific surface protein markers, which may depend on many factors, such as their origin. In the case of exosomes, some proteins tendentially common among them and often used in the identification of exosomes include annexin, CD9, CD63, CD81, HSP70, and flotillin [92,93].

It has been recognized that EVs play a fundamental role in intercellular communication, functioning as vehicles for transporting and delivering a range of cellular bioactive cargos, including membrane and cytosolic proteins, lipids, DNA, mRNA (messenger RNA), and miRNA (microRNA) [94,95]. EVs may directly influence the cellular state of recipient cells through their specific shuttled contents. This occurs via miRNA-induced gene expression posttranscriptional regulation [96], which includes numerous cellular epigenetic regulations [97,98]. EVs play a role in the maintenance of cellular homeostasis by being pivotal to cellular uptake mechanisms [99]. An example of this is the ligand/receptor interaction within brain synaptic transmission [100,101]. EVs are also important in the maintenance of stem cell plasticity [102] and in the formation of new tissues, as they are important for angiogenesis [103,104], the generation of morphologic gradients for tissue genesis during neuronal development [105,106], and neuronal regeneration [107,108].

Regarding their release pathways, EVs are delivered into the extracellular space via the SNARE-mediated fusion of multivesicular endosomes with the plasma membrane [109]. The direct budding of vesicles with the plasma membrane results in microvesicles [110,111]. Additionally, vesicles that may be shed from cells undergoing programmed cell death originate from apoptotic bodies [112]. Following exocytosis, EVs may remain in the extracellular space surrounding the secreter cell or, instead, travel elsewhere, such as in the brain, by crossing the blood–brain barrier (BBB) [113], or from the brain into the periphery. Within the brain, exosomes are released by several cell types, including neurons [114], microglial cells [115], astrocytes [116], and oligodendrocytes [117].

Different EVs are noticeable in plasma [118], urine [119], breast milk [120], cerebrospinal fluid [121], semen, peritoneal and bronchoalveolar lavage fluids, amniotic fluid, and even tumor effusions [83,86], thus allowing long-distance intercellular information exchange [89].

### 2.2. The Role of EVs in ALS

EVs have been associated with numerous pathologies, from metastatic cancers [122] to neurodegenerative diseases such as Alzheimer’s disease and Parkinson’s disease [123,124,125]. Under such pathological conditions, EVs shuttle enclosed misfolded proteins and other neurotoxic elements that can potentially induce dysfunction in recipient cells [125,126]. EVs are increasingly recognized as being of great importance in the pathogenesis of ALS and in the identification of biomarkers, which will be explored in this section (Figure 1).

#### 2.2.1. EVs in ALS Disease Progression and Pathological Mechanisms

EVs have emerged as significant players in ALS progression, with increasing evidence pointing to their role in the dissemination of detrimental biocargo. EVs allow for the hypothetical prion-like propagation of ALS-related mutant misfolded proteins and dysregulated miRNAs [81], which are believed to contribute to disease severity and progression [127,128,129]. The most common cargos found in EVs from patients with ALS include misfolded proteins such as mSOD1, FUS, TDP43, and C9orf72 expansions DPRs and other neurotoxic elements [64,81]. These harmful cargos have been screened in both astrocytes and neuron-derived exosomes in different ALS disease models, such as the SOD1-G93A mouse model, which is one of the most commonly used animal models for studying ALS. In this model, the mutated SOD1 gene harbors a glycine-to-alanine substitution at codon 93. Recently, ref. [130] demonstrated that mutant SOD1 (mSOD1) accumulation occurs in cellular vacuoles, which may be constituted by different portions of organelles, and once released, leads to the existence of different types of EVs, particularly mitoEVs. The formation and type of vacuoles and the resulting EVs appear to be related to the stage of ALS pathology in this mouse model. Interestingly, before the onset of motor symptoms, these vacuoles are already present and are mainly of mitochondrial origin, with a high content of mSOD1, ultimately resulting in the release of mSOD1-containing EVs [130]. The authors of this study hypothesized that these EVs, derived from damaged neurons, may be responsible for the initiation of a sequence of signaling cascades that contribute to neuroinflammation, glial-mediated neurotoxicity, and prion-like spreading of the disease. The muscle-specific expression of mutant SOD1(G93A) can also have a negative effect on neurons, as it has been reported to alter and dismantle neuromuscular junctions through a PKCθ-dependent mechanism [131]. The expression of mutant SOD1 induces the upregulation of PKCθ, and its colocalization with acetylcholine receptors (AChR) leads to a decrease in mitochondrial function, alterations in redox signaling, and neuromuscular junctions hindering transmission [131]. Moreover, and as noted in Peggion et al. [132], myocytes from hSOD1(G93A) mice are susceptible to reactive oxygen species that lead to an unbalanced mitochondrial redox state and changes in Ca^2+^ homeostasis. This ultimately triggers a reactive glial response and the release of proinflammatory cytokines that affect both motor neurons and neuromuscular junctions (NMJs). To support this finding, cytokines such as IL-1b, IFN-y, and IL-6 were found in circulating EVs of the spinal cord from SOD1(G93A) mice [133]. The existence of different vacuole/EV phenotypes and associated cell death pathways may have different roles in the onset and severity of symptoms, as well as in the heterogeneity and progression of the disease.

Exosomal TDP-43 is another significant cargo that plays a crucial role in ALS progression. A longitudinal study conducted on ALS patients demonstrated an increase in the exosomal TDP-43 ratio in peripheral blood during the course of the disease, particularly in the early stages [134]. This increase in the TDP-43 ratio is associated with elevated levels of neurofilament light chain (NFL) in the plasma of these patients, which is more prevalent in individuals with rapid disease progression [134]. Further evidence supports the significance of exosomal TDP-43 in the disease propagation. Ding et al. (2015) described the damaging effect of exosomes enclosing TDP-43 C-terminal fragments (CTFs) from the cerebrospinal fluid of ALS patients with FTD (ALS-FTD-CSF) in human glioma cells (U251 cells). After incubation with ALS-FTD-CSF-derived exosomes, naive U251 cells developed intracellular TDP-43 aggregates in the form of tunneling nanotube (TNTs)-like structures [135]. Although in vivo studies are required, this previous work suggests that EVs may act as vehicles for the spread of TDP-43 aggregates in the context of ALS.

EVs and their toxic payloads not only damage neurons but also spread pathological signaling by transferring them between different cell types, including neurons, astrocytes, and muscle cells. Evidence of these interactions was provided by a study showing that the EV-mediated transfer of DRPs occurred between MNs-like NSC34 cells and rat cortical neurons and, then, from those to rat cortical astrocytes [64]. This transfer is relevant to ALS, as EVs carrying C9orf72-encoded DPRs were identified to be involved in the exchange between human C9orf72-induced pluripotent stem cell-derived motor neurons (hiPSC-MNs) and control iPSC-derived spinal MNs [64]. In NSC34 cells transfected with mutant SOD1(G93A) (hSOD1-G93A NSC34 cells), miR-124 was found to be upregulated and transferred to EVs. When these cells were cocultured with N9-microglial cells, miR-124 contained in mSOD1 exosomes was translocated to N9-microglial cells, resulting in phenotypic alterations, such as a reduction in their phagocytic capability and activation of neuroinflammation pathways [136]. Exosomes released by mouse astrocytes overexpressing G93A SOD1 were also previously shown to be responsible for the transfer of mutant SOD1 to mouse spinal neurons and to induce MN death [137]. Moreover, astrocytic-derived exosomes from the plasma of sALS patients were found to transport inflammation-related cargo, including IL-6, a proinflammatory interleukin, which was increased in these vesicles and positively associated with the rate of disease progression [138]. The negative impact of EVs and their cargo on the interaction of affected muscle cells with MNs was further demonstrated by evidence that multivesicular bodies released from ALS muscle cells are neurotoxic to healthy MNs [139]. In this study, EVs derived from muscle cells obtained from biopsies of sALS patients were exposed to healthy hiPSC-MNs and were shown to be neurotoxic through increased FUS expression, resulting in shorter and less branched neurites, atrophic myotubes, and enhanced cell death [139]. The observed cell death was greatly reduced by immunoblocking the vesicle uptake by MNs with anti-CD63. Finally, a study by Anakor et al. supported the cause and effect relationship between muscle cell vesicles and MNs. The exposure of MNs to skeletal muscle cell-derived exosome-like vesicles (MuVs) in ALS patients resulted in reduced neurite length, number of neurite branches, and reduced MN survival and myotubes by 31% and 18%, respectively. Moreover, adding ALS-derived MuVs to healthy astrocytes led to an increase in the proportion of stellate astrocytes and, thus, mild activation of these cells [140].

#### 2.2.2. miRNAs and Misfolded Proteins EVs Cargo in ALS: Potential ALS Biomarkers

One particular cargo of EV miRNAs has attracted research interest as potential biomarkers for ALS due to their versatile functions in regulating gene expression across a wide range of processes, including neural development, cell proliferation and differentiation, protein ubiquitination, apoptosis, and other transcriptional regulatory processes (summarized in Table 1). Despite their link to ALS progression, the mechanisms underlying the alterations in their expression and levels remain inconclusive. Defective RNA metabolism and miRNA dysregulation are closely associated with ALS [141]. miRNA profiles in ALS exhibit significant variations among patients and can be over- or under-expressed as they are transported by EVs across multiple biofluids and tissues (summarized in Table 1). Most of the research has focused on screening plasma circulating EVs of ALS patients using a variety of research methodologies, ranging from RT-qPCR analysis to microarrays [142].

In the quest for an ALS molecular biomarker fingerprint, ref. [158] reported the downregulation of miR-27a-3p in serum-derived exosomes from patients with ALS. Saucier and colleagues [145] found 27 differentially expressed miRNAs, 5 of which were upregulated and 22 downregulated when compared via next-generation sequencing, the EVs isolated from plasma samples of ALS patients compared to those from healthy controls. Some miRNAs were relevant to ALS diagnosis, as they were related to the Revised ALS Functional Rating Scale (ALSFRS-R) scores. This was the case for miR-193a-5p, which allowed us to distinguish between patients with low and high scores. miR-15a-5p has been shown to be important in differentiating controls from patients with ALS. In a separate study, Katsu and coworkers [153] analyzed miRNA profiles in neuron-derived EVs from plasma samples of ALS patients via microarrays and identified 30 differentially expressed miRNAs: 13 upregulated and 17 downregulated. In another study, Pregnolato et al. [272] performed miRNA screening of serum-derived exosomes using RT-qPCR analysis. Owing to the small sample size used in this study, no statistically significant differences were observed in the expression levels of any miRNA. However, a recent study by Lo et al. [155] analyzed the miRNA cargo profiles of EVs isolated from postmortem homogenates of the frontal cortex, spinal cord, and serum of patients with sALS. The authors found no difference in the number of EVs between patients and controls, but patients with ALS presented larger spinal cord vesicles and smaller-sized vesicles in the serum. Two miRNAs related to axon guidance and long-term potentiation were significantly dysregulated in all analyzed tissues: miR-342-3p was upregulated, and miR-1254 was downregulated. Furthermore, the miRNA levels were reduced in the frontal cortex and spinal cord of sALS patients, whereas they were increased in the serum. Another study, performed by Rizzuti et al. [162], analyzed EVs isolated from MN cultures obtained from fibroblast-reprogrammed iPSCs of ALS patients carrying *C9ORFf72*, *SOD1*, and *TARDBP* mutations. These authors found the dysregulation of several miRNAs, specifically the upregulation of miR-629-5p and miR-194-5p and downregulation of miR-34a-5p, miR-1267, and miR-625-3p. Interestingly, the latter was found to be consistently downregulated in C9orf72 MN exosomes and upregulated in EVs from TARDBP-MNs. In the same study, miR-625-3p was also predicted to mediate cell-to-cell communication, immune system pathways, and autophagy. Furthermore, in another study by the same authors [273] using iPSC-derived MNs progenitors from fALS and sALS patients, further dysregulation was found, notably of miR-34a, which is involved in cell cycle regulation, autophagy, apoptosis, neurogenesis, and neuronal differentiation [274]. Sproviero et al. (2021) also searched for potential ALS EV miRNA biomarkers and found the dysregulation of hsa-miR-206, hsa-miR-205-5p, miR-1-3p, hsa-miR-205-5p, hsa-miR-200b-3p, hsa-miR-200c-3p, hsa-miR-6888-3p, hsa-miR-31-5p, hsa-miR-141-3p, and hsa-miR-210-3p in the plasma of ALS patients [275]. Using an in situ hybridization analysis, Yelick et al. (2020) found the downregulation of miR-124-3p in exosomes from SOD1-G93A mice spinal MNs. Moreover, in this study, we found a significant correlation between cerebrospinal fluid (CSF) exosomal miR-124-3p expression levels and the disease stage of male ALS patients, as denoted by the ALSFRS-R score [149]. It is worth noting that miR-124-3p is a recognized oncogene [276,277] with an essential role in cell proliferation and apoptosis [276] and is associated with poor survival rates in patients with hepatocellular carcinoma [278]. Conversely, its upregulation was shown to decrease the metastatic behavior of hepatocarcinoma cells through the reversion of CRKL expression, which resulted in the suppression of the extracellular signal-regulated kinase (ERK) pathway and inhibition of malignant cell proliferation [279]. Importantly, its upregulation was found to be protective against post-traumatic neurodegeneration through activation of the Rela/ApoE signaling pathway [280], and its downregulation was linked to the neurodegeneration and neuroinflammatory states of post-traumatic brain injuries (TBI) [281].

Other miRNAs that were differentially expressed in serum-derived extracellular vesicles from 50 patients with ALS were reported recently by [150]. Statistically significant robust results yielded a differential expression of seven miRNAs included in extracellular vesicles, two of which were upregulated (miR-151a-5p and miR-146a-5p) and three downregulated (miR-4454, miR-10b-5p, and miR-29b-3p) [150]. Among the reported functions, these specific miRNAs have been associated with tumorigenesis [282,283,284] and protection against cell apoptosis [285].

Despite recent advances in the understanding of the role of miRNAs associated with EVs in driving the progression of ALS, this field is still in its early stages. An analysis of miRNA expression profiles suggests that the current knowledge is insufficient to predict their involvement in the pathological mechanisms of ALS [286]. In a study that analyzed the results of research from 2013 to 2018, Foggin et al. (2019) reported that most dysregulated miRNAs were either upregulated or downregulated. This outcome may be due to intrinsic differences in the methodologies used for miRNA detection or other factors, such as differences in miRNA expression across different tissues and sample extraction protocols. Interestingly, eight of the nine most commonly dysregulated miRNAs were predicted to target at least one of the most commonly mutated genes in ALS, but a random sample of unrelated miRNAs that were not found to be dysregulated in ALS patients also yielded a similar prediction [286]. Nonetheless, the search for miRNAs as potential biomarkers for ALS remains promising because of their good preservation in different types of biological samples, such as CSF and blood, often with an advantage over several proteins in allowing for a more reliable and faster diagnosis and closer classification and understanding of each case. In this scope, and as suggested by Rizzuti and colleagues [162], it is important to analyze miRNAs isolated from different human biological samples (e.g., MNs, exosomes, and CSF) in different ALS types. Likewise, miR-206 has been proposed as a potential biomarker in a study by Toivonen et al. (2014), since it displayed consistent changes towards its upregulation in ALS disease progression in SOD1 mice [287]. miR-206 is a microRNA that has been identified as a tumor suppressor involved in regulating the transforming growth factor-β (TGF-β) signaling pathway [288]. Importantly, it is one of the canonical myomiR due to its high expression in skeletal muscle [289], where it is involved in myogenesis and skeletal tissue regeneration [290,291,292]. In several studies related to ALS pathology, consistent expression levels of miR-206 have been observed across different biological samples. For example, miR-206 was found to be overexpressed in the serum of sALS patients [293] and in both plasma and skeletal muscles of spinal onset ALS patients [294]. In a study performed with the SOD1-G93A ALS mouse model, miR-206 overexpression was found to be associated with the onset of neurological symptoms, which may be attributed to skeletal muscle denervation [290]. Indeed, the downregulation of miR-206 restored neuromuscular synapses, indicating the potential of miR-206 as a therapeutic target for ALS [290]. In a recent study performed on SOD1-G93A mice, the pivotal role of miR-206 and miR-133a in skeletal muscle remodeling was evidenced [292]. This skeletal muscle remodeling by miR-133a seems to be related to myoblast proliferation by inhibiting the serum response factor (SRF) [295]. The study reported a significant decrease in the levels of both miR-206 and miR-133a in the serum of these animals 2 and 10 days after surgical-induced nerve dissection. However, after 30 days post-surgery, the miR-206 levels returned to normal, indicating its critical involvement in skeletal muscle reinnervation. Furthermore, the study found that the miR-206 expression levels in the serum could serve as an indicator of ALS disease progression, as a significant upregulation of this miRNA was observed during the late symptomatic phase of ALS (at 220 days). It is worth noting that miR-206 has been detected in EVs in ALS [275], suggesting its potential release into the bloodstream and contribution to the disease progression.

In addition to miRNAs, the protein cargo of EVs associated with ALS may also hold potential as a novel biomarker (summarized in Table 2). In a study by Vassileff and colleagues [296], 12 proteins were identified as being exclusive to EVs derived from the postmortem motor cortex tissue of ALS patients, including CD177, CHMP4B, CSPG5, DYNC1I2, IGHV3-43, LBP, RPS29, S100A9, SAA1, SCAMP4, SCN2B, and SLC16A1. Additionally, Pasetto and collaborators [297] discovered a potential new method for patient stratification based on the levels of cyclophilin A, a protein involved in TDP-43 trafficking and function, in combination with the EV size distribution in plasma-derived EVs from ALS patients. This approach can be used to distinguish between slow and fast disease progression. Recently, Sjoqvist and Otake [298] conducted a pilot study comparing CSF and CSF-EVs from patients with ALS and matched control subjects to search for novel ALS biomarker candidates. They found four differentially expressed proteins in the CSF of ALS patients, including downregulated MB and upregulated JAM-A, TNF-R2, and CHIT1. Although no proteins were differentially expressed in CSF-EVs, there was a trend for the downregulation of perlecan, a proteoglycan of the extracellular matrix involved in cell proliferation, differentiation, adhesion, migration, and tissue repair and regeneration [299]. Conversely, Thompson et al. (2020) found no significant differences in CSF-EV concentration and size distribution between the control and ALS groups. However, they identified altered protein homeostatic mechanisms in ALS patients, including the downregulation of bleomycin hydrolase [300], a cytosolic cysteine protease that has been associated with the release of chemokines in inflammation and wound healing processes [301]. These data, together with those indicating the involvement of EVs in aggregated protein spreading, suggest that the analysis of EV protein contents is mandatory for the development of innovative diagnostic and prognostic tools and for the identification of new therapeutic targets for ALS. Overall, while these studies have provided promising results, further research is needed to understand the role of miRNAs and proteins transported by EVs in ALS development and progression and their possible use as biomarkers.

## 3. Therapeutic Perspectives with EVs in ALS

In recent years, several studies have proposed innovative next-generation EV-related therapies that hold great promise for the treatment of human diseases [329]. EVs present several therapeutic advantages owing to their higher biocompatibility and reduced immunogenicity compared to alternative carriers, such as synthetic nanocarriers, which may also be prone to accumulation in the liver and spleen [330,331]. As natural nanoparticles, EVs can be easily isolated from various biofluids and can cross biological barriers to deliver potential therapeutics [332]. Despite some uncertainty regarding their functional mechanisms, it is becoming clear that these bioparticles shuttle diverse cargos capable of recapitulating the benefits of “whole-cell therapy”, either by preventing or mitigating abnormal cellular functions [333].

One emerging therapeutic approach related to EVs is the use of stem cell-derived EVs. These EVs have a positive impact on the pathophysiology of various neurodegenerative diseases [334]. In ALS, these EVs may achieve beneficial effects by modulating the immune system, addressing mitochondrial dysfunction, and boosting neuroprotection in MNs [335,336,337]. For example, exosomes derived from adipose-derived stem cells (ASCs) obtained from SOD1-G93A mice were shown to have a neuroprotective effect by reducing oxidative stress-related damage in MN-like NSC-34 cells overexpressing ALS-associated mutations, including SOD1(G93A), SOD1(G37R), and SOD1(A4V) [336]. Furthermore, the same research group observed that NSC-34(G93A) cells internalized ASC-derived exosomes, leading to the downregulation of proapoptotic proteins (Bax and cleaved caspase-3) and the upregulation of antiapoptotic proteins (Bcl-2α), ultimately improving neuronal survival [338]. In a more recent study, ASC-derived exosomes obtained from SOD1-G93A mice were used to slow the progression of ALS by reducing glial cell activation and improving motor performance. Interestingly, these exosomes showed an affinity towards the lesioned areas of the brain, suggesting targeted delivery, although the exact mechanisms behind this phenomenon require further elucidation [339]. Similarly, ASC-derived exosomes were found to increase the expression levels of phospho-CREB/CREB and PGC-1α in neurons derived from the neural stem cells of SOD1-G93A mice. This results in a reduction in cytosolic SOD1 aggregates and rescues mitochondrial dysfunction [318]. Additionally, the same type of exosomes was shown to rescue the inherent impairment in oxidative phosphorylation (OXPHOS) specifically linked to mitochondrial complex I in NSC-34(G93A) cells [340]. In their study, human bone marrow endothelial progenitor cell (hBM-EPC)-derived exosomes were shown to restore mouse brain endothelial cells previously damaged through in vitro exposure to SOD1-G93A mutant male mouse plasma. These results indicated a significant reduction in microvascular endothelial damage. Interestingly, blocking the β1 integrin of exosomes using an anti-CD29 blocking antibody prevented their internalization by recipient cells, thereby increasing the percentage of brain endothelial cell death. These findings suggest that hBM-EPC-derived exosomes have the potential to repair endothelial damage in ALS and that their internalization by recipient cells may play a critical role in their therapeutic effects. Garbuzova-Davis et al. (2020) investigated the potential therapeutic role of exosomes derived from human hBM-EPCs in the repair of endothelial damage in ALS. To induce damage, researchers exposed a mouse brain endothelial cell line to plasma from SOD1-G93A male mice. They found that plasma-derived exosomes treatment significantly increased endothelial cell death. However, a significant reduction in cell death was obtained by supplementing the brain endothelium, previously exposed to ALS plasma-derived exosomes, with 1 μg/mL of hBM-EPC-derived exosomes for 24 h. Moreover, when these EVs were pretreated with an anti-CD29 blocking antibody to block β1 integrin, they were prevented from being internalized by recipient cells, resulting in a significant increase in brain endothelial cell death. These findings suggest that hBM-EPC-derived exosomes have the potential to reduce the number of damaged endothelial cells in ALS, but their beneficial effects may be dependent on proper cellular internalization [341]. In contrast, the negative effects of ALS-related EVs were reversed by Varcianna et al. (2019). The authors isolated EVs from human-induced astrocytes derived from C9ORF72-ALS sALS patients (C9ORF72-ALS iAstrocyte-derived EVs) and found that they originally compromised both neurite network maintenance and MN survival in HB9-GFP+ mouse-cultured MNs (Hb9-GFP + MNs). This effect was related to the downregulation of the miR-494-3p content in these EVs. Nevertheless, following treatment with C9ORF72-ALS iAstrocyte-derived EVs, where the miR-494-3p levels were intentionally upregulated, HB9-GFP + mouse-cultured motor neurons presented neurite network restoration and decreased MN death. These beneficial effects of miR-494-3p overexpression may be related to its function as a negative regulator of semaphorin 3A (SEMA3A) and other targets involved in axonal maintenance [205].

In addition to the described therapeutic possibilities, EVs have emerged as promising drug carriers with the potential to deliver synthetic drugs to the brain. This is especially important because many proteic and small-molecule neurological drugs may fail to bypass the blood–brain barrier (BBB), which can hinder their effectiveness [342]. The encapsulation of these drugs within EVs could help overcome this limitation by allowing them to cross the BBB and improve drug targeting and efficiency [343]. While EVs are not currently used to deliver drugs for ALS treatment, they have been employed in the treatment of other brain diseases, such as brain tumors, using doxorubicin-loaded exosomes [344]. Therefore, this approach may also be a promising avenue for future ALS research.

Although previous findings demonstrated promise for the potential application of EV-based therapies in ALS, it is important to acknowledge the existing challenges and limitations in this field, which align with the obstacles encountered in ALS research. The intricate nature of ALS, coupled with the substantial heterogeneity among patients, adds to the complexity. Moreover, obtaining patient samples that involve brain and nervous tissue poses invasive methodologies, making it arduous. Consequently, researchers resort to animal models or in vitro systems. However, these alternatives have inherent limitations and fail to capture the full disease complexity and phenotypic variations across the entire population, thereby lacking translation to real clinical scenarios. Innovative approaches involving EVs, like genome therapy [345] or immunomodulation, hold the potential to establish more robust foundations for advancing ALS therapeutics. An intriguing avenue for investigation lies in harnessing EVs in conjunction with CRISPR/CAS9 gene editing technologies [346,347]. It is worth noting that, to the best of our knowledge, no clinical studies have been conducted or proposed involving the use of EV-based therapies in ALS with human patients, which demonstrates that research in this field is strongly needed.

## 4. Conclusions

ALS is a fatal neurodegenerative disease with a complex and unclear etiopathology that strongly affects patient health and well-being. With no cure available thus far, searching for an effective treatment that can improve patients’ life expectancy and quality of life is paramount. ALS presents several important challenges and hurdles to the research because of the intrinsic complexity and heterogeneity of the disease. Nevertheless, important advances have been made in recent years. Among these are the recent advances in ALS-related EV research, which are emerging as key players in the surfacing and development of the disease by allowing for the transport of biomolecular cargo from cell to cell, thus spreading anomalies across the system. EVs also have the potential to be employed as a source of potential biomarkers for the early detection of ALS and personalized prognostic purposes. Furthermore, they may also be exploitable to tackle existing altered mechanisms and for applications in a variety of therapeutic strategies, such as being employed for drug delivery, as they can carry different types of molecules, both natural and artificial. Specifically, within the ALS research area, stem cell-derived EV use is increasing for therapeutic purposes, which is of higher relevance given the disease heterogeneity and allows for a precision-based approach.

While the use of stem cell-derived EVs for therapeutic purposes is promising, further innovative and consensual approaches are needed to reverse the disease’s biopathologic mechanisms and translate this knowledge into real-life applications that can bring hope to both patients and their families.

## Figures and Tables

**Figure 1 cells-12-01763-f001:**
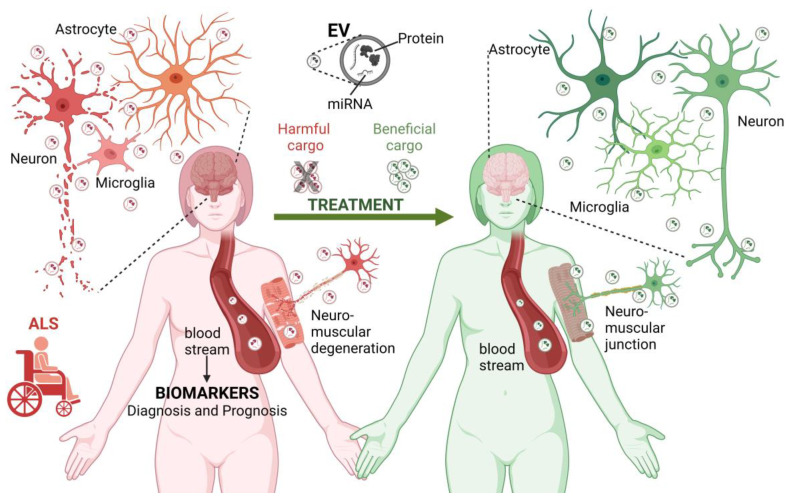
The role of EVs in ALS. EVs contribute to the pathogenesis of ALS (**left reddish side**). EVs are produced by different cell types in the central nervous system and neuromuscular junctions. In the context of ALS, EVs may carry disease-related biological molecules (proteins and miRNAs) involved in the transformation and degeneration of the brain and neuromuscular elements, thus contributing to the spread of the pathology between different cell types. Furthermore, EVs can reach long distances in the body, contributing to the exchange of harmful molecules between the brain and neuromuscular junction. Considering this, the molecules transported by EVs circulating in the bloodstream and cerebrospinal fluid are considered potential biomarkers for the diagnosis and prognosis of ALS. Finally, EVs have a therapeutic potential. Blocking the exchange of EVs carrying harmful molecules and administering EVs with neuroprotective cargo may slow the progression of ALS or revert its pathological effects (**right-greenish side**). ALS, amyotrophic lateral sclerosis, EV, extracellular vesicle; miRNA, microRNA. Figure created using BioRender.com (accessed on 6 June 2023).

**Table 1 cells-12-01763-t001:** miRNA cargo in amyotrophic lateral sclerosis-related extracellular vesicles.

miRNA	Main Targets	Biological Role	Expression
**miR-9-1-5p**, **miR-9-2-5p**, **miR-9-3-5p**	*PAK4*, *CoREST*, *CPEB3*, *ECAD*, *elavl3*, *FoxG1*, *Hes1*, *IGF2BP3*, *Nr2e1/TLX*, *REST*, *Sirt1*, *Zic5*; IGF2-PI3K/Akt signaling	Cell differentiation regulation, neuronal function, synaptic plasticity neurotransmitter release; skeletal muscle cell proliferation, and differentiation inhibition regulation [143]; apoptosis inhibition [144]	**downregulated**: blood plasma-derived EVs [145]
**miR-10b-5p**	*NFAT5*; KLF11-KIT signaling	Regulation of insulin production, lipid metabolism and gastrointestinal motility [146], and myoblasts differentiation [147]. Tumorigenic inhibitor [148]	**downregulated**: CSF exosomes and blood plasma-derived EVs [149,150]
**miR-15a-5p**	*BCL2*, *Cyclin D1*, *FEAT*, *PD-1*, *ROR1*, CXCL10-ERK-LIN28a-let-7 axis, NF-κB signaling, Wtn/β-catenin signaling	Tumor progression inhibition [151]	**upregulated**: blood plasma-derived EVs [145]
**miR-24-3p**	*eNOS*, *GATA2*, *PAK4*	Tumor suppression, angiogenesis regulation, and cell protection against apoptosis [152]	**upregulated**: blood plasma-derived EVs [153]
**miR-26a-5p**	*ADAM17*, *Bid*, *FAF1*, *SERBP1*, *Wnt5*; TGF-β signaling	Osteogenic differentiation and cell proliferation regulation [154]	**upregulated**: serum [155]
**miR-27a-3p**	*AQP11*, *BTG2*	Tumor suppression [156] and protection against the blood–brain barrier and brain injury after brain hemorrhage [157]	**downregulated**: serum-derived exosomes [158]
**miR-29b-3p**	C1QTNF6/AMPK signaling	Modulation of inflammatory response [159]	**downregulated**: CSF exosomes and blood plasma-derived EVs [149,150]
**miR-34a**	*AXIN2*, *BCL2*, *BIRC5*, *CD44*, *DGKζ*, *E2F3*, *MDMX*, *MET*; *MYCN*, *NOTCH1*, *NANOG*, *PD-L1*, *SIRT1*, *SNAI1*, *SOX2*; cyclins, cyclin-dependent kinases, TGF-β1/Smad signaling	Cell proliferation, apoptosis, autophagy and cellular senescence regulation [160,161], and matrix proteins deposition [161]	**downregulated**: ALS iMNs-derived exosomes and CSF [162]
**miR-100-5p**	*ANKAR*, *AP1AR*, *EPDR1*, *ICK*, *NR1I3*, *SMARCA5*, *ST6GALNAC4*, *TMPRSS13*, *TTC39A*; mTOR signaling	Cell survival regulation (e.g., apoptosis) [163] and autophagy [164,165]	**downregulated**: blood plasma-derived EVs [145]
**miR-124–3p**	*CDK6*, *EfnB1*, *PTBP1*, *REST*, *SCP1*, *Sox9*; *NeuroD1*;	Synaptic connectivity and plasticity regulation [166]	**upregulated**: CSF exosomes [149]
**miR-127-3p**	*BCCIP*, *BOLA1*, *FAM27D1*, *KCNK2*, *LOC100134822*, *MTCP1*, *PSD95*, *RBPMS*, *SIRT3*, *SLC25A2*, *TPTE2*, *ZNF3*, NeuroD1, NR2A-subunit	Neurogenesis, synapse formation and motor neuron integrity maintenance, mitophagy, ROS, and misfolded proteins accumulation [167]	**upregulated**: serum [155]; downregulated: blood plasma [145,155]
**miR-144-3p**	*ABCA1*, *CCNT2*, *FoxO1*, *FST*, *GABRA1*, *HGF*, *IGIP*, *NFE2L2*, *ST3GAL6*, *UBE2D1*, *UBR3*	Adipogenesis regulation, metastasis, and cell proliferation inhibition [168]	**upregulated**: blood plasma-derived EVs [145]
**miR-146a-5p**	*IRAK-1*; NF-κB signaling	Immune cell activity, hematopoiesis, and malignant transformation regulation [169]	**upregulated**: blood plasma-derived EVs [150,170]
**miR-149-3p**	*AKT2*	Cell proliferation inhibition in cancer [171]	**upregulated** in blood plasma-derived EVs [153]
**miR-150-3p**	*CASP2*, *SP1*	Neuroprotection of neural stem cells exosomes after ischemic insult and cell proliferation inhibition [172,173]	**downregulated**: blood plasma-derived EVs [153]
**miR-151a-3p**	*SOCS5*, *SP3*, *YTHDF3*; JAK2/STAT3 signaling	Tumorigenic inhibitor [174]	**upregulated**: blood plasma-derived EVs [150,170]
**miR-151a-5p**	*AGMAT*, *CYTB*, *SMARCA5*	Cellular ATP production regulation [175]	**upregulated**: blood plasma-derived EVs [150,170]
**miR-181a-1-5p**	*Kras*, *NRAS*, *VCAM-1*, *ZNF780A*, *ZNF780B*, *ZNF204P*, *ZNF439*, *ZNF527*, *ZNF559*, *ZNF594*, *ZNF781*, *ZNF844*	Tumorigenic suppressor, immune response regulation, and cell proliferation [176,177,178]	**downregulated**: blood plasma-derived EVs [145]
**miR-181a-2-5p**	*STAT3*, *TGFβR3*	Tumorigenic suppressor [179]	**downregulated**: blood plasma-derived EVs [145]
**miR-181b-1-5p**	*BAZ2B*, *NOVA1*, *TGFβ1*, *ZNF780A*, *ZNF780B*, *ZNF439*, *ZNF527*, *ZNF559*, *ZNF594*, *ZNF781*, *ZNF844*; MEK/ERK/p21 pathway	Cell proliferation, invasion and metastasis in cancer [180], apoptosis inhibition [181], and autophagy [182]	**downregulated**: blood plasma-derived EVs [145]
**miR-181b-2-5p**	*BCL2*, *TIMP3*; annexin A2	Cell migratory proteins modulation [183] and chemosensitivity in cancer cells [184]	**downregulated**: blood plasma-derived EVs [145]
**miR-183-5p**	*AKAP12*, *CCDC121*, *DHRSX*, *FKSG83*, *GNG5*, *NUDT4*, *PFN2*, *PDCD4*, *PSEN2*, *RIPK3*, *SLAIN1*, *XPNPEP3*,	Neuron protection against motor cell death in ALS (under stress conditions) [185]	**upregulated**: blood plasma-derived EVs [145]
**miR-194-5p**	*HIF-1*, *NR2F2*, *NR2F6*, *PAK2*; MAPK1/PTEN/AKT signaling	Tumorigenic inhibition [186]	**upregulated**: ALS iMNs-derived exosomes [145,162]
**miR-197-3p**	*HSPA5*, *KIAA0101*, *TIMP2/3*; AKT/mTOR axis signaling	Tumor suppression, cell proliferation [187], autophagy regulation [188], and angiogenesis promotion [189]. Recognized biomarker for myocardial fibrosis and heart failure [190]	**downregulated**: postmortem frontal cortex and spinal cord [155]
**miR-199a-1-3p**	*BCAR3*, *CDNF*, *DNMT3a*, *FABP12*, *HVCN1*, *KLHL3*, *RAP2a*; *SERPINE2SRR*, *TMEM161B*, *TSGA10*, *WFDC8*	Growth and angiogenesis inhibition in tumors [191]	**downregulated**: blood plasma-derived EVs [145]
**miR-199a-2-3p**	caveolin-2, Ppargc1a, Sirt1	Regulation of cell proliferation and survival [192]	**downregulated**: blood plasma-derived EVs [145]
**miR-199a-3p**	*CCND1*, *CD44*, *c-MYC*, *DNMT3a*, *EGFR*, *ETNK1*, *YAP1*; mTOR	Cell proliferation regulation and apoptosis induction [192]	**upregulated**: blood plasma-derived EVs [150,170]
**miR-199b-3p**	*CDNF*, *BCAR3*, *FABP12*, *HVCN1*, *KLHL3*, *SERPINE2 TSGA10*, *SRR*, *TMEM161B*, *WFDC8*; *Phospholipase Cε*	Tumor suppression [193]	**downregulated**: blood plasma-derived EVs [145]
**miR-199a-5p**	*DDK1*, *ITGA3*, *WTN2*; CREB/BDNF signaling, NF-κB signaling	Tumorigenic inhibitor [194]	**upregulated**: blood plasma-derived EVs [150,170]
**miR-298**	*JMJD6*	Tumor suppression, cell proliferation, and metastasis inhibition [195]	**downregulated**: postmortem frontal cortex, spinal cord and serum [155]
**miR-335**	*ROCK1*, survivin	Tumor suppression [196]	**downregulated**: ALS iMNs-derived exosomes [162]
**miR-338-3p**	*C5ORF47*, *C6ORF141*, *DGKB*, *IDNK PREX2*, *IZUMO3*, *PIM1*, *ROBO1*, *SP3*, *TAX1BP3*, *ZNF141*, *ZNF208*	Tumor suppression; cell proliferation, migration, and invasion inhibition [197,198]	**downregulated**: blood plasma-derived EVs [145]
**miR-342-3p**	*ATF3*, *FOXQ1*, *RAP2B*, *MAP1LC3B*; HDAC7/PTEN axis signaling, RhoC GTPase	Prion-based neurodegeneration and intracellular motor proteins, axon guidance, cell proliferation and apoptosis regulation [199,200], tumor suppression, autophagy, and reduction of cell stemness [201]	**upregulated**: postmortem frontal cortex, spinal cord and serum-derived EVs [155]
**miR-363-3p**	*CD69*, *DCAF6*, *FAM24A*, *FBXW7*, *FNIP1*, *MAN2A1*, *FBXW7*, *KLF4*, *PTEN*; PI3K/AKT signaling	Osteogenic differentiation [202]	**upregulated**: blood plasma-derived EVs [145]
**miR-371a-5p**	*BCL2*; *BECN1*, *SOX2*	Tumor suppression; cell proliferation, migration, and autophagy [203]	**upregulated** in blood plasma-derived EVs [153]
**miR-450a-2-3p**	*FOXP3*, *IGF2*, *MAPK1*, *KSR2*	Tumorigenic inhibition [204]	**upregulated**: postmortem spinal cord and serum [155]
**miR-494-3p**	*SEMA3A*	Axonal maintenance (negative regulation of semaphorin 3A (SEM3A)) [205]	**downregulated**: astrocyte-derived EVs and in cortico-spinal tract tissue [205]
**miR-502-5p**	*SP1*	Tumor suppression, regulation of cell proliferation, and migration [206]	**downregulated**: postmortem frontal cortex and spinal cord [155]
**miR-512-5p**	*ETS1*, *hTERT*, *SOD2*; Wnt/β-catenin signaling	Tumor suppression and apoptosis induction [207]	**upregulated**: postmortem frontal cortex [155]
**miR-520f-3p**	*C2orf69*, *NDST4*, *SOX9*, Wnt signaling	Tumor suppression [208]	**upregulated**: serum [155]
**miR-532-3p**	*C13orf34*, *C22orf46*, *DNAL4*, *ENSA*, *FOXP3*, *KLHL12*; *OPHN1*, *RPRML*, *RPS3*, *ZNF514*; β-catenin	Cell proliferation, metastasis inhibition, and apoptosis enhancing [209]	**upregulated**: blood plasma-derived EVs [145]
**miR-551b-3p**	*H6PD*, Cyclin D1, TRIM31/Akt signaling	Tumor inhibition [210]	**upregulated**: serum [155]
**miR-549a**	yet to be studied	Angiogenesis and metastasis induction [211]	**downregulated**: postmortem frontal cortex and spinal cord [155]
**miR-587**	*RPSA*	Tumor suppression [212]	**downregulated**: serum-derived EVs [155]
**miR-625-3p**	*GABBR2*, *SCAI*	Cancer cells migration and invasion inhibition [213]	**downregulated**: ALS iMNs-derived exosomes and CSF [162]
**miR-629-5p**	*AKAP13*, *CAV1*, *SFRP2*	Tumor cell growth regulation [214]	**upregulated**: ALS iMNs- derived exosomes [145,162]
**miR-634**	*HSPA2*; mTOR signaling	Tumor suppression and apoptosis enhancing [215,216]	**downregulated**: blood plasma-derived EVs [153]
**miR-664a-5p**	*AC093802.1*, *ANKRD36*; *CCNDBP1*, *DNASE2*, *FBXO17*, *HMGA2*, *IDH2*, *PTCD3*, *SEPT7*, *ZNF256*, *ZNF772*	Osteogenic differentiation, controlled apoptosis [217], and neuronal differentiation [218]	**downregulated**: blood plasma-derived EVs [145]
**miR-766-3p**	NF-κB signaling, TGFBI signaling	Inhibition of inflammatory responses [219] and apoptosis promotion in cancer [220]	**downregulated**: serum-derived EVs [155]
**miR-877-5p**	*FOXM1*	Tumor suppression, cell proliferation, migration, and invasion reduction [221]	**downregulated**: serum-derived EVs [155]
**miR-939-5p**	*ARHGAP4*, *HIF-1 alpha*, *IGF-1R*; PI3K/Akt signaling	Cell migration and invasion in certain types of cancer [222]	**upregulated**: blood plasma-derived EVs [153]
**miR-1207-5p**	*CX3CR1*; NF-κB signaling, SARS-CoV-2 RNA	inflammatory response regulation [223]	**upregulated**: blood plasma-derived EVs [153]
**miR-1246**	*CDR1as*, *DNAH*, *FAM53C*, *FAM169B*, *GSG1L*, *KIAA1370*, *LIG4*; *NFE2L3*, *NR2F2*, *SGOL1*, *WDR77*, *ZNF23*, *ZNF267*; NHEJ signaling	Modulation of DNA damage following ionizing radiation exposure [224,225]	**upregulated**: blood plasma-derived EVs [145]
**miR-1254**	*Smurf1*; PIK/Akt signaling,	Cell proliferation, migration, and invasion inhibition [226]	**downregulated**: postmortem frontal cortex, spinal cord, and serum [155]
**miR-1255a**	*SMAD4*; TGF-β signaling	Related with breast cancer malignant phenotype and downstream effector of TGF-β [227]	**upregulated** in serum [155]
**miR-1260b**	*C2orf48*, *CASP8*, *CTAGE1*, *GOLGA8A*, *MED13L*, *PABPN1*, *USP48*, *ZNF256*, *ZNF594*, *ZNF788*; MAPK pathway	Tumorigenesis promotion [228]	**downregulated**: blood plasma-derived EVs [145]
**miR-1262**	*SCL2A1*, *ULK1*	Tumor suppression [229]	**upregulated** in serum [155]
**miR-1268a**	*ABCC1*	Mediation of temozolomide resistance in glioblastoma [230]	**downregulated**: blood plasma-derived EVs [153]
**miR-1268b**	*AKT*, *BCL2*, *ERBB2*, *PI3KCA*, PI3K-AKT signaling	Apoptosis inhibition [231]	**Upregulated**: serum [155]
**miR-1285-5p**	*CDH1*, *Smad4*, *TMEM194A*	Cell proliferation regulation [232]	**upregulated**: postmortem frontal cortex [155]
**miR-1290**	*AKAP7*, *CDR1as*, *FAM19A5*, *HIGD2A*, *OGN MYO10*, *OSBPL6*, *RP11-1167A19.2*, *SGOL1*, *WDR77*	Cell proliferation, migration, and invasion regulation in cancer [233]	**downregulated**: blood plasma-derived EVs [145]
**miR-1913**	not yet studied, but 732 predicted targets in [234]	Potential noninvasive biomarker for prostate cancer [235]	**downregulated**: blood plasma-derived EVs [153]
**miR-2861**	*STAT3*, *MMP2*, EGFR/AKT2/CCND1 signaling	Tumor suppression, cell proliferation regulation, and apoptosis [236]	**downregulated**: blood plasma-derived EVs [153]
**miR-3176**	*AR*, *PTEN*	Promotion of tumorigenesis and tumor progression [237]	**downregulated**: blood plasma-derived EVs [153]
**miR-3177-3p**	not yet studied, but 65 predicted targets in [234]	to be studied	**downregulated**: blood plasma-derived EVs [153]
**miR-3605-5p**	*SCABR2*	Adipocyte lipolysis regulation [238]	**downregulated**: blood plasma-derived EVs [153]
**miR-3619-3p**	Wnt/β-catenin signaling	Cell migration and invasion promotion [239]	**upregulated**: blood plasma-derived EVs [153]
**miR-3911**	not yet studied	Possible sALS biomarker [142]	**downregulated**: blood plasma-derived EVs [153]
**miR-3940-3p**	*BIRC5*, *IL-2Ry*, *KCNA5*, Integrin α6	Regulation of maternal insulin resistance, T-cell activity promotion, and metastasis inhibition in cancers [240]	**downregulated**: blood plasma-derived EVs [153]
**miR-4286**	*APLN*, *C15orf34*, *CBX2*, *FAM222B*, *HKDC1*, *INPP4A*, *ZFP36L1*, *PARVG*, *PRX PTEN*, *RNF43*, *SALL1*; *TMSB4X*, JAK2/STAT3 signaling, PI3K/Akt signaling, TGF-ß/TGF-ß1/Smad signaling	Cell proliferation, apoptosis, and inflammatory response modulation [241,242,243,244]	**downregulated**: blood plasma-derived EVs [145]
**miR-4298**	*SOD2*, *TGIF2*	Cell proliferation, migration, and invasion of cancer cells [245]	**upregulated**: blood plasma-derived EVs [153]
**miR-4443**	*INPP4A*, *METLL3*, *TRIM14*; JAK2/STAT3 signaling, NF-κB signaling, Ras signaling, TGF-β1 signaling	Metastasis and energy metabolism suppression [246]	**downregulated**: postmortem frontal cortex and spinal cord [155]
**miR-4454**	*ABHD2/NUDT21*, *Vps4a*, *Rab27A*; GNL3L/NF-κB signaling; TGF-β/MAPK pathway	Insulin signaling [247], metastasis progression in cancer [248,249,250] and apoptosis regulation [251]	**downregulated**: CSF exosomes and blood plasma-derived EVs [145,149,150]; **upregulated**: serum [155]
**miR-4505**	*HSPA12B*	Tumorigenesis [252] vascular function [253]	**upregulated**: blood plasma-derived EVs [153]
**miR-4507**	*TP53*; PI3K-AKT signaling	Cell proliferation and migration in lung cancer [254]	**downregulated**: blood plasma-derived EVs [153]
**miR-4508**	*125 predicted in miRDB* [234]	Potential involvement in pulmonary fibrosis through an unknown mechanism [255]	**downregulated**: blood plasma-derived EVs [153]
**miR-4646-5p**	*ABHD16A*; *PHD3*; PHD3/HIF1A signaling	Ubiquitination and cell proliferation and invasion regulation [256,257]	**downregulated**: blood plasma-derived EVs [153]
**miR-4674**	p38k	Angiogenesis regulation [258]	**downregulated**: blood plasma-derived EVs [153]
**miR-4687-5p**	*ATP10D*, *THRSP*	Involved in polycystic ovary syndrome [259]	**downregulated**: blood plasma-derived EVs [153]
**miR-4688**	not yet studied	miRNA sponge and cancer progression [260]	**upregulated** blood plasma-derived EVs [153]
**miR-4700-5p**	not yet studied	not yet studied	**upregulated** blood plasma-derived EVs [153]
**miR-4736**	*AR*	Inflammatory response regulation [261]	**upregulated** blood plasma-derived EVs [153]
**miR-4739**	*BMP-7*; ITGA10/PI3K signaling	Apoptosis and differentiation regulation [262]	**upregulated** blood plasma-derived EVs [153]
**miR-4745-5p**	SIRT6/PCSk9 signaling	sensibility to anesthetics regulation [263]	**upregulated** blood plasma-derived EVs [153]
**miR-4788**	not yet studied, but 29 predicted targets in miRDB [234]	Nervous system development, neurotransmitter levels regulation and transport, and synapsis [264]	**downregulated**: blood plasma-derived EVs [153]
**miR-7641-1**	*BCL2*, *CAS9*, *C9orf153*, *PARP*, *RAB7L1*, *RPS16*, *TMEM156*, *TMPRSS11BNL*, *TNFSF10*	Apoptotic signaling in cancer [265]	**downregulated**: blood plasma-derived EVs [145]
**miR-7975**	*MTDH*	Possible involvement in atherosclerosis [266], in lung inflammation, and cancer [267,268]	**upregulated**: serum of sALS patients [155]
**miR-7977**	*CD84*, *MRPS12*, *MRPL27*, *MUC19*, *TRAPPC2*, *SIRT3*, Hyppo-YAP signaling	Hematopoiesis regulation [269], oxidative stress, and insulin resistance [270]	**downregulated**: blood plasma-derived EVs [145]
**let-7c-5p**	*ARID3B*, *C14orf28*, *DNA2*, *FIGN*, *HMGA2*, *LIN28B*, *TRIM71 SMARCAD1*; CTHRC1/AKT/ERK signaling	Microglia activation inhibition. Cell migration and proliferation. Inhibition and apoptosis enhancement [271]	**downregulated**: blood plasma-derived EVs [145]

**Table 2 cells-12-01763-t002:** Protein contents in ALS-derived extracellular vesicles.

Protein Content	Biological Function	Vesicle/Sample Type	Main Results
**BLMH**	Enzyme with proteolytic activity. Involved in release of inflammatory chemokines and in wound healing [301]	**EVs** from CSF of ALS patients (C9orf72 mutation)	Downregulation in ALS patients CSF-derived EVs
**CD163, FOXP3, IL2RA, MRC1**	Anti-inflammatory transcripts [302]	Treg-derived **EVs** from spinal cord from SOD1^G93A^ mice model and iPSC-derived from myeloid cells	- **Upregulation** in spleen-derived myeloid cells after Treg- derived EVs treatment [133]; - Intranasal administration of enriched Treg EVs slowed disease progression, increased survival, and modulated inflammation within the SOD1^G93A^ mice spinal cord [133]
**CD177, CHMP4B, CSPG5, DYNC1I2, IGHV3-43, LBP, RPS29, S100A9, SAA1, SCAMP4, SCN2B, SLC16A1, STAU1, FXYD6, DYNC1I1, DHX30**	Involvement in stress granule dynamics [296]	**MCEVs** from ALS patients’ postmortem motor cortex tissue	- 12 RNA-binding proteins only found in MCEVs from ALS patients (mainly **downregulated**) [296]; - 4 proteins significantly **upregulated** in MCEVs from ALS patients (DYNC1I1, DHX30, FXYD6, STAU1) [296]
**CHIT1**	Cleavage of chitin (protein found in cell walls of various pathogens). Expressed during the later stages of macrophage differentiation. Important in inflammation and tissue remodeling. In the ALS context, plays a role in the feed-forward loop that maintains inflammation [303]	**EVs** from ALS patients’ CSF	**Upregulation** in ALS patients CSF-derived EVs [298]
**CUEDC2**	Regulates ubiquitin-proteasome pathways and inflammatory response [304]	**Exos** from sALS patients’ CSF	**Only expressed in ALS** group [305]
**FUS and pFUS**	DNA repair, RNA splicing, dendritic RNA transport, miRNA function and biogenesis [306]	**MVs** and **Exos** from sALS patients’ plasma	Protein levels are **higher in ALS** patients’ plasma-derived MVs than Exos [81]
**HSP90**	Chaperone protein involved in protein folding [307]	**EVs** from ALS patients and symptomatic SOD1G93A and TDP-43Q331K ALS mice models plasma	**Downregulation** in EVs from sALS patients [297]
**JAM-A**	Regulation of several processes including paracellular permeability, platelet activation, angiogenesis and the modulation of junctional tightness in the blood brain barrier (BBB) [308]	**EVs** from ALS patients CSF	**Downregulation** in ALS patients CSF-derived EVs [298]
**IL-6, iNOS, IL-1b, IFN-y**	Proinflammatory cytokines (IL-1b, IFN-y, IL-6) and enzyme (iNOS) produced in response to cytokines [309]	Treg-derived **EVs** from spinal cord from SOD1^G93A^ mice model and iPSC-derived from myeloid cells	**Downregulation** in spleen-derived myeloid cells after Treg-derived EVs treatment [133]
**MB**	Oxygen-binding molecule expressed in skeletal and cardiac muscle tissue [310,311]	**EVs** from ALS patients CSF	**Downregulation** in ALS patients CSF-derived EVs [298]
**mfSOD1**	Antioxidant enzyme, protects cells from ROS [312]	**Vacuoles** and **EVs** from spinal cord samples of SOD1^G93A^ mice model	**Accumulation** of mfSOD1-vacuoles in degenerating MNs, released into the extracellular space in the form of extracellular vesicles [130]
**NIR**	Translocates from the nucleolus to the nucleoplasm in response to the nucleolar stress [313]	**Exos** from sALS patients CSF and anterior horn postmortem tissue sections	**Upregulation** in sALS patients CSF-derived exos [314]
**Nrf2**	Antioxidant factor [315]	**EVs** from spinal cord tissue of SOD1^G93A^ mice model	**Upregulation** after exposure to MSCs-derived EVs, with consequent reduction of ROS [316]
**pCREB**	Involved in the synthesis of proteins required for LTP [317]	**Exos** from SOD1^G93A^ mice model SVZ-derived NSCs, differentiated into G93A neuronal cells	**Downregulation** in G93A cells, normalized with ADSC-exos treatment [318]
**PGC-1α**	Involved in the regulation cell metabolism [319]	**Exos** from SOD1^G93A^ mice model SVZ-derived NSCs, differentiated into G93A neuronal cells	**Downregulation** levels in G93A cells, normalized with ADSC-exos treatment [318]
**Phenylalanine**	Precursor for tyrosine [320], the monoamine neurotransmitters dopamine, norepinephrine, and epinephrine	**lEVs** and **sEVs** from sALS patients’ plasma	**Downregulation** in EVs from sALS patients [321]
**pMLKL**	Effector of necroptotic pathways [322]	**Vacuoles** and **EVs** from spinal cord samples of SOD1^G93A^ mice model	**Upregulation** in vacuoles of degenerating MNs (necroptotic pathway activation/phenotype 3) [130]
**PPIA**	Ubiquitous protein involved in protein folding, transport and signaling (e.g apoptosis, inflammation, etc) [323]	**EVs** from ALS patients and symptomatic SOD1G93A and TDP-43Q331K ALS mice models plasma	Protein levels and EV size distribution distinguish fast and slow ALS disease progression [297]
**pNFH**	Chaperone involved in TDP-43 trafficking and function [324]	**EVs** from ALS patients and symptomatic SOD1G93A and TDP-43Q331K ALS mice models plasma	**Upregulation** in EVs from sALS patients [297]
**SOD1**	Binds copper and zinc ions, responsible for freeing superoxide radicals from cells [312]	**Exos** from SOD1^G93A^ mice model SVZ-derived NSCs, differentiated into G93A neuronal cells	ADSC-exos alleviated aggregation of cytosolic SOD1 in G93A ALS mice isolated neuronal cells [318]
		**MVs** and **Exos** from sALS patients’ plasma	Protein levels are **higher** in ALS patients’ plasma-derived Exos than MVs [81]
**TDP-43 and pTDP-43**	RNA regulation (transcriptional regulation, alternative splicing and mRNA stabilization) [325]	**Exos** from ALS patients CSF	TDP-43 **accumulation** [326]
		**MVs** and Exos from sALS patients’ plasma	Protein levels are **higher** in ALS patients’ plasma-derived MVs than Exos [81]
**TNF-R2**	Proinflammatory proteins activation [327]	**EVs** from ALS patients’ CSF	**Upregulation** in serum of ALS patients. TNF-R2 knocking down in ALS mouse model results in motor neuron protection [298]
**UBA1**	Involved in ubiquitination of proteins for degradation by the UPS [328]	**EVs** from ALS patients’ CSF (C9orf72 mutation)	**Upregulation** in ALS patients’ CSF-derived EVs [298]

Abbreviations: ALS (amyotrophic lateral sclerosis); ADSC (exo-adipose-derived stem cell exosomes); BLMH (pentameric proteasome-like protein bleomycin hydrolase); CD206 (MRC1; mannose receptor C-Type 1); CHIT1 (chitotriosidase/chitinase 1); CHMP4B (charged multivesicular body protein 4B); CSPG5 (chondroitin sulfate proteoglycan 5); CSF (cerebrospinal fluid); CSF-EVs (CSF-derived extracellular vesicles); CUEDC2 (CUE domain-containing protein 2); DHX30 (DExH-box helicase 30); DYNC1I1 (dynein cytoplasmic 1 intermediate chain 1); DYNC1I2 (dynein cytoplasmic 1 intermediate chain 2); Exos (exosomes); EVs (extracellular vesicles); FOXP3 (forkhead box P3); FUS (fused in sarcoma); p-FUS (phosphorylated fused in sarcoma); FXYD6 (FXYD domain-containing ion transport regulator 6); lEVs (large extracellular vesicles); IFN-y (interferon-γ); IL-1β (interleukin-1β); IL-2RA (interleukin-2 receptor subunit α); IL-6 (interleukin-6); IGHV3-43 (immunoglobulin heavy variable 3-43); IGHV3-43 (immunoglobulin heavy variable 3-43); iNOS (inducible nitric oxide synthase); JAM-A (junctional adhesion molecule-A); LBP (lipopolysaccharide-binding protein); MB (myoglobin); MCEVs (motor cortex extracellular vesicles); mfSOD1 (misfolded protein SOD1); MNs (motor neurons); MRC1 (mannose receptor C-type 1/CD206); MSCs (mesenchymal stem cells); MVs (microvesicles); NIR (INHAT repressor); Nrf2 (nuclear factor E2-related factor 2); NSCs (neuronal stem cells); pCREB (phosphorylated cAMP response element-binding protein); PGC-1α (peroxisome proliferator-activated receptor-γ coactivator); pMLKL (phosphorylated mixed lineage kinase domain-like protein); pNFH (phosphorylated neurofilament protein heavy unit); PPIA (cyclophilin A); ROS (reactive oxygen species); RPS29 (ribosomal protein S29); sALS (sporadic amyotrophic lateral sclerosis); sEVs (small extracellular vesicles); SCAMP4 (secretory carrier membrane protein 4); SCN2B (sodium channel subunit β-2); SLC16A1 (solute carrier family 16 member 1); SOD1 (superoxide dismutase 1); STAU1 (Staufen double-stranded RNA-binding protein 1); SVZ (subventricular zone); S100A9 (S100 calcium-binding protein A9); TNF-R2 (tumor necrosis factor receptor 2); TDP-43 (TAR DNA-binding protein 43); Treg (regulatory T cells); UBA1 (ubiquitin-activating enzyme E); UPS (ubiquitin-proteasome system).

## Data Availability

No new data were created or analyzed in this study. Data sharing is not applicable to this article.

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
