# Peer review of "Recent Advances in Extracellular Vesicles in Amyotrophic Lateral Sclerosis and Emergent Perspectives"

_cells, 2023, doi:10.3390/cells12131763_

Round 1
Reviewer 1 Report
The review is quite good, comprehensive, and relevant in the current context, as EVs are gaining substantial traction in diagnostics and disease development. The main strength lies in its coverage of topics that make it more exciting. It is English that authors need to look at and revise. Other comments I made that are for authors are mainly suggestions, and it is up to them to consider.
Title: Recent Advances in extracellular vesicles in Amyotrophic Lat-2 eral Sclerosis and emergent perspectives, by Gonçalo J.M. Afonso and colleagues.
Comments: This review report focuses on the emerging role of extracellular vesicles (EVs) in amyotrophic lateral sclerosis (ALS), a fatal and complex neurodegenerative disease. ALS poses significant challenges to research due to its heterogeneity and unclear etiopathology. However, recent advances in ALS-related EVs research have highlighted their involvement in the disease's development and progression. EVs facilitate the transport of biomolecular cargo between cells, contributing to the spread of anomalies throughout the system. Additionally, EVs are promising biomarkers for early ALS detection and personalized prognostic purposes.
Furthermore, EVs have the potential to be utilized in therapeutic strategies, including drug delivery, due to their ability to carry various types of molecules. Particularly, stem cell-derived EVs are gaining attention for their potential therapeutic applications, offering a precision-based approach to tackle the disease's heterogeneity. However, further innovative and consensus-driven approaches are required to unravel the pathological mechanisms of ALS and translate this knowledge into real-life applications that bring hope to patients and their families.
Importance of the Review:
This review report is of utmost importance as it addresses the urgent need for effective treatments for ALS, a devastating disease without a cure. By highlighting the significant advancements in ALS-related EVs research, the review emphasizes the potential of EVs as critical players in disease development and progression. Exploring EVs as biomarkers holds promise for early diagnosis and personalized prognosis, which could significantly impact patient outcomes. Moreover, the review underscores the therapeutic potential of EVs, mainly stem cell-derived EVs, offering a precision-based approach to address the heterogeneity of ALS. By acknowledging the challenges that lie ahead, the review inspires the scientific community to develop innovative and consensus-driven approaches to translate EV-based knowledge into tangible applications that can bring hope to ALS patients and their families.
Possible Improvements:
While the review provides a compelling overview of the promising role of EVs in ALS research and therapy, there are areas where improvements can be made. Firstly, the review could provide more specific details about the types of biomolecular cargo carried by EVs in ALS and their relevance to disease mechanisms. Additionally, discussing the current limitations and challenges associated with EV-based approaches in ALS research and therapy would provide a more comprehensive perspective. Furthermore, exploring the potential of EVs in other therapeutic modalities beyond drug delivery, such as immunomodulation or gene therapy, could expand the scope of the review. Lastly, incorporating insights into ongoing clinical trials or potential future directions would enhance the practical applicability of the review's findings.
Overall, this review report highlights the promising role of EVs in ALS research and therapy, emphasizing their potential as biomarkers and therapeutic agents. With the suggested improvements, this review has the potential to serve as a valuable resource for researchers, clinicians, and stakeholders involved in the pursuit of effective treatments for ALS.
I would strongly recommend authors run their paper by Paperpal so that the English language is further improved. In many places, the English language appears skewed, and authors need to adopt the proper language usage in the scientific context.
Author Response
The authors express their gratitude to the reviewer for their valuable comments and revision of the manuscript. As suggested, we ran the manuscript through the Paperpal tool and incorporated the recommended alterations to enhance the clarity of the English language. These modifications were implemented throughout the document but were not made visible to the reviewers, as they did not impact the content of the revision. Making them visible could potentially confuse the second revision process. Additionally, we included a paragraph (page 17, line 592) to draw the reader's attention to the topic of the role of extracellular vesicles (EVs) in immunomodulation and gene therapy, as well as the lack of clinical trials in this area. However, due to the length of the review, we were unable to delve into these aspects in detail. All the changes made in the manuscript are indicated in blue. As a result of addressing the topic of myomiRNA and its significance in skeletal muscle cells in response to reviewer 2, new references have been added. Consequently, the reference numbers have been updated from that page onwards.
Reviewer 2 Report
I find this review complete both from the point of view of the analysis of the genes involved, the role of EVs and miRNAs in ALS. However, I think it is necessary to better address the emerging role of other cell type in ALS onset and progression, taking in mind of the Cell Autonomous and Non-Cell Autonomous mechanisms. Have authors some idea whether known alterations in skeletal muscle (physiology, secretion or whatever else) can act on the motor neuron , for example by releasing toxic factors, including myomiRNA or some cytokines?
Here some articles regarding such aspects (PMID: 34440812; PMID: 28931313; PMID: 35453299; PMID: 27679581)
Author Response
The authors express their gratitude to the reviewer for their valuable comments and revision of the manuscript.
Reviewer comment:
Have authors some idea whether known alterations in skeletal muscle (physiology, secretion or whatever else) can act on the motor neuron , for example by releasing toxic factors, including myomiRNA or some cytokines?Here some articles regarding such aspects (PMID: 34440812; PMID: 28931313; PMID: 35453299; PMID: 27679581)
Authors response:
In order to enhance our review and delve into the role of extracellular vesicles (EVs) in skeletal muscle alterations, we have introduced this topic on page 7, lines 286-300, and page 13, lines 442-463. We appreciate the suggestion of articles related to this subject and have incorporated them into this newly added section of the text. All the modifications made in the manuscript are highlighted in blue. As a result of the addition of new references, the reference numbers have been updated from page 7 onwards.
Following the suggestion of reviewer 1 we ran the manuscript through the Paperpal tool and incorporated the recommended alterations to enhance the clarity of the English language. These modifications were implemented throughout the document but were not made visible to the reviewers, as they did not impact the content of the revision. Making them visible could potentially confuse the second revision process.